# Protective Effect of Dinitrosyl Iron Complexes Bound with Hemoglobin on Oxidative Modification by Peroxynitrite

**DOI:** 10.3390/ijms222413649

**Published:** 2021-12-20

**Authors:** Olga V. Kosmachevskaya, Elvira I. Nasybullina, Konstantin B. Shumaev, Natalia N. Novikova, Alexey F. Topunov

**Affiliations:** 1Research Center of Biotechnology of the Russian Academy of Sciences, Bach Institute of Biochemistry, 119071 Moscow, Russia; rizobium@yandex.ru (O.V.K.); lvirus198709@rambler.ru (E.I.N.); tomorov@mail.ru (K.B.S.); 2National Research Center, Kurchatov Institute, 123182 Moscow, Russia; nn-novikova07@yandex.ru

**Keywords:** hemoglobin, oxidative modification, dinitrosyl iron complexes, peroxynitrite, carbonyl derivatives

## Abstract

Dinitrosyl iron complexes (DNICs) are a physiological form of nitric oxide (^•^NO) in an organism. They are able not only to deposit and transport ^•^NO, but are also to act as antioxidant and antiradical agents. However, the mechanics of hemoglobin-bound DNICs (Hb-DNICs) protecting Hb against peroxynitrite-caused, mediated oxidative modification have not yet been scrutinized. Through EPR spectroscopy we show that Hb-DNICs are destroyed under the peroxynitrite action in a dose-dependent manner. At the same time, DNICs inhibit the oxidation of tryptophan and tyrosine residues and formation of carbonyl derivatives. They also prevent the formation of covalent crosslinks between Hb subunits and degradation of a heme group. These effects can arise from the oxoferryl heme form being reduced, and they can be connected with the ability of DNICs to directly intercept peroxynitrite and free radicals, which emerge due to its homolysis. These data show that DNICs may ensure protection from myocardial ischemia.

## 1. Introduction

Dinitrosyl iron complexes (DNICs) are a physiological form of nitric oxide (^•^NO) in an organism, depositing ^•^NO and transferring it to various biological targets [1,2,3]. These complexes get formed in humans, animals, plants, and bacteria. DNICs have a wide range of biological activities that are connected with ^•^NO, and NO^+^ is released during decay of the complexes. Along with ^•^NO, thiol-containing compounds, such as cysteine and glutathione, usually act as ligands of low-molecular DNICs in living systems [4,5]. DNICs donate ^•^NO and can also be the donors of Fe–(NO)_2_ group, and participate in nitrosylation of thiol SH-groups. It should be noticed that as nitric oxide is a radical molecule, we indicate it as ^•^NO, although very often it is written simply: NO.

Low molecular weight DNICs with thiol ligands have a common formula:

(RS^−^)_2_–Fe^+^–(NO^+^)_2_, and they always exist in an equilibrium with protein-bound DNICs in an organism. The latter are formed with thiol and some other protein groups [2,5]. This equilibrium is maintained via redox conditions, mostly by the state of the glutathione system. A decrease in the reduced glutathione (GSH) pool shifts the balance towards more stable protein-bound DNICs.

Experimental models both in vitro and in vivo revealed that NO can act as an antioxidant in reactions with radicals (O_2_^•^^−^, ^•^OH, Tyr^•^, RS^•^, ^•^NO_2_, ROO^•^). Since nitrosylated and nitrated lipids are formed in the reaction with ROO^•^, ^•^NO can terminate chain reactions of lipid peroxidation [6]. Meanwhile, ^•^NO can also act as a pro-oxidant, being a source of reactive nitrogen species. The most cytotoxic ^•^NO derivative is peroxynitrite (ONOO^−^) [7], formed in the diffusion-controlled reaction of ^•^NO with a superoxide [8]:
^•^NO + O_2_^•^^−^ ⟶ ONOO^−^    (k ≈ 10^10^ M^−1^s^−1^)(1)

Peroxynitrite can also be formed in the reaction of the triplet nitroxyl anion NO^−^ with molecular oxygen [9]:NO^−^ + O_2_ ⟶ ONOO^−^    (k = 2.7 × 10^9^ M^−1^s^−1^)(2)

Toxic ONOO^−^ properties are largely determined by hydroxyl radicals and nitrogen dioxide formed during its decomposition:ONOO^−^ + H^+^ ⟶ HOONO ⟶ ^•^OH + NO_2_^•^(3)

Production of peroxynitrite increases in case of pathologies and metabolic disfunctions including but not limited to inflammation, cardiovascular, and neurodegenerative diseases [10]. In these cases, peroxynitrite mainly arises from macrophages, the cells of the immune system, which produce large amounts of ONOO^−^ in response to pathogen invasion [11]. 

^•^NO penetrates red blood cells (RBC) by free diffusion and can also be produced in RBC during the reduction of nitrite ions by deoxyHb [12]. If ^•^NO interacts with Hb in RBC, it competes with the ^•^NO reaction with superoxide (O_2_^•^^−^) [11]. Superoxide can form in RBC as well. This can occur under hypoxic conditions, when hemoglobin (Hb) rapidly releases oxygen at low pO_2_ [13].

DNICs can demonstrate both antioxidant properties, connected with NO action, and pro-oxidant ones, conditioned by peroxynitrite. We have previously shown that low molecular weight and protein-bound DNICs intercept superoxide anion radicals, and DNICs bound with Hb (Hb-DNICs) protect its constituent thiol groups against oxidation [14,15]. In those experiments, hydrogen peroxide and *tert*-butyl hydroperoxide were used as oxidation initiators. 

Hemoglobin (Hb) was the object of this study, since this protein is the main target of peroxynitrite action in RBC [16]. In addition, Hb is one of the proteins that DNICs bind to [14,15]. Figure 1 shows a schematic image of DNIC bound with tetrameric erythrocytic Hb. One of the ligands of this complex is Cys93 residue of the Hb β-subunit, the other may be an anionic group of amino acid residue spatially close to Cys93. This amino acid residue is denoted as R^−^.

Antioxidant effect of protein-bound DNICs is still poorly understood, as contrasted with low molecular weight DNICs with thiol ligands, although the comprehension of the mechanisms of their action is very important for the solving of both medical and biotechnological issues. Therefore, the study of Hb-bound DNICs and their antioxidant properties was the main purpose of this work. 

## 2. Results

### 2.1. Destruction of Hb-DNICs with Peroxynitrite Action, Using EPR Spectroscopy

Low-molecular-weight and protein-bound DNICs are destroyed by various oxidizing agents (peroxides and superoxide anion radicals) [14,15,17,18]. Therefore, we investigated the destruction of metHb-bound DNICs (Hb-DNICs) under various ONOO^−^ concentrations. 

Figure 2A shows the electron paramagnetic resonance (EPR) spectra of Hb-DNICs. These spectra are characterized by high axial symmetry and the g-factor of 2.03. It should be noted that g = 2.03 signals are a distinctive feature of DNICs containing thiol ligands [2]. It is also known that Cys93 residues of β-subunits act as thiol ligands in Hb [14,15].

Hb-DNICs were quantitatively destroyed by peroxynitrite at physiological pH (Figure 2). The kinetics of DNICs degradation with ONOO^−^ action has two phases (Figure 2B). The first phase of rapid Hb-DNICs destruction lasts for 4–5 min and is probably associated with the interaction of complexes with peroxynitrite and its reactive derivatives (Figure 2B). If the molar ratio of MetHb to ONOO^−^ is 1:12 (ONOO^−^ concentration = 4.8 mM), the amount of Hb-DNICs decreases by ~75% (Figure 2A), and with the ratio of 1:14.5 (ONOO^−^ concentration = 5.8 mM), paramagnetic DNICs are almost completely destroyed.

During the second (slower) phase, the rate of Hb-DNICs decay in the control and in the presence of peroxynitrite was almost the same (Figure 2B). Apparently, the destruction of DNICs in this case was caused by oxidation with molecular O_2_, as it was shown in [19]. However, when Hb-DNICs are destroyed by peroxynitrite, the EPR signal characteristic of nitrosyl complexes with heme iron is not formed, in contrast to their destruction by superoxide [14,15]. 

### 2.2. Formation of Hb Carbonyl Derivatives

The formation of carbonyl derivatives is one of the main oxidative modifications of biomolecules, i.e., proteins. In our experiments, DNICs had a pronounced inhibitory effect on metHb carbonylation in the presence of peroxynitrite. They protected Hb in a wide range of ONOO^−^ concentrations (0.38–4.20 mM) (Figure 3). With metHb:ONOO ratio of 1:6.5, the amount of Hb-DNICs carbonyl derivatives was 60% lower than in a control sample without DNICs (Figure 3, curves 1 and 2).

The curves showing the dependence of the amount of carbonyl derivatives on oxidant concentration, had a clear two-phase character in the presence of DNICs: with ONOO^−^ concentration of 1.6 mM the carbonyl amount was insignificant, while with the concentration exceeding 1.6 mM, the oxidative metHb modification was more intensive. This effect was probably related to the release of iron from decayed DNICs, involved in further generation of free radicals in Fenton-type reactions. This is confirmed by the data on the effect of peroxynitrite on DNICs destruction (point 1 of this section), showing that with this protein/oxidant ratio, a large portion of complexes are destroyed.

Variable valence metals cause decomposition of the hydroperoxides of amino acid residues with alkoxyl radical (RO^•^) formation [20,21]:Fe^2+^ + ROOH → Fe^3+^ + RO^•^ + OH^−^(4)

These metals catalyze site-specific protein oxidation, including the formation of carbonyl products [20,22]. The effect of dopant iron on ONOO^−^-dependent protein oxidation also cannot be excluded. This effect of variable valence metals was shown in the oxidation and nitration of protein tyrosine residues [16,23]. 

To verify these assumptions, the experiments were performed in the presence of a chelator (DTPA), which forms a complex with free iron ions. This complex does not take part in the Fenton and Haber–Weiss reactions. DTPA adding to the reaction mixture containing metHb and peroxynitrite significantly inhibited the carbonyls’ formation (Figure 3, curves 3 and 4). It indicates a significant contribution of dopant iron to the oxidative modification of the protein.

Hb can also serve as a potential source of iron ions in the reaction system, since, during Hb oxidative modification, the degradation of the heme group takes place [24].

### 2.3. Degradation of Hb Heme Group

The main target of peroxynitrite action on metHb is the heme group [25]. In our experiments, protein-bound DNICs inhibited destruction of the heme group (Figure 4A) while in the control samples, the portion of destroyed heme directly depended on the concentration of the oxidizer (Figure 4A, curve 1). During the oxidation of the heme group with peroxynitrite, ferryl and oxoferryl heme forms are formed, which degrade to biliverdin with the release of iron. Oxidative degradation of biliverdin leads to the generation of a number of fluorescent products. With ONOO^−^ concentration = 4.2 mM, 50% of heme were destroyed, while only 25% were destroyed in Hb-DNICs (Figure 4A, curve 2). This can be related to the fact that under the impact of peroxynitrite, β-subunits of Hb are primarily destroyed [26].

In the metHb reaction with peroxynitrite, the ferric–peroxynitrite complex (Hb-Fe^III^ONOO^−^) is formed [16,27]. Subsequent changes of this complex may be different. If the transient ferric-peroxynitrite complex is formed, during subsequent rearrangement it turns into ferric-nitrate species, and then is decomposed to nitrate and ferrylHb [28]:Hb-Fe^III^-OONO^−^ → [Hb-Fe^III^-NO_3_^−^] → Hb-Fe^III^ + NO_3_^−^(5)

Thus, Hb catalyzes peroxynitrite isomerization to nitrate [29]. 

In another scenario, the homolysis of peroxynitrite O-O bond leads to decomposition of the ferric−peroxynitrite complex and formation of oxoferrylHb and nitrogen dioxide [30]:
[Hb-Fe^III^-OONO^−^]^2+^ → Hb-[Fe^III^-O▪▪ONO]^2+^→ [Hb-Fe^IV^=O]^2+^ + ^•^NO_2_(6)

Several mechanisms of the protective DNICs effect can be suggested. Firstly, DNICs are ^•^NO donors, which can reduce Hb-Fe^IV^=O. In this reaction ferric–nitrite species (Hb-Fe^III^-ONO complex) are formed [17]. Secondly, DNICs can protect the heme group by transferring ^•^NO to heme iron, which prevents the formation of the oxoferrylHb form [31,32,33,34]. Thirdly, the molecular mechanism of DNICs antioxidant action may be related to the reduction of radicals of the oxoferryl porphyrin form [17,31,33]. The DNICs’ ability to directly interact with organic free radicals was shown in our works [17,35]. Finally, the inclusion of iron ions released during heme decay into DNICs may possibly prevent them from taking part in the peroxynitrite homolytic decomposition catalysis with free radical product formation. 

It is known that under peroxide impact, Hb can transform into a hemichrome state. In this case heme iron forms a coordination bond with the distal histidine residue (bis-His hexacoordinated species) [36]. In the hemichrome state Hb is prone to the formation of insoluble macromolecular aggregates. Here we observed Hb conversion to hemichrome, starting with ONOO^−^ concentration 1:10.6 (1.6 mM) (Figure 4B). This was proved by the absence of 500 and 631 nm peaks, characteristic of the high-spin state of a heme iron (Fe^3+^-H_2_O), and the simultaneous appearance of a 536 nm peak and a 565 nm shoulder, inherent for the low-spin iron form (His-Fe^3+^-His) [37]. 

### 2.4. Oxidation of Tyrosine and Tryptophan Residues in Hb 

In Hb interaction with peroxynitrite, free radicals of tyrosine, tryptophan, and cysteine are produced. The oxoferryl heme [38,39,40], hydroxyl radical and nitrogen dioxide [39,41] act as oxidants here. In this reaction of one-electron oxidation of tyrosine residues, phenoxyl radicals are formed [39,42]. The changes of the tyrosine phenoxyl radical may be different. The reaction of two phenoxyl radicals produces 3,3′-dityrosine dimer [43], but in the reaction of tyrosine radicals with a ferric-peroxynitrite complex and ^•^NO_2_, 3-nitrotyrosine is formed [44,45]. In our experiment DNICs were ^•^NO donors, and the presence of metal-binding centers in the protein accelerates this reaction [23,46]. In addition, the tyrosine radical can react with ^•^NO, forming 3-nitrosotyrosine, which is subsequently transformed into 3-nitrothyrosine [39]. The reactions listed above, leading to free radical modification of tyrosine residues in proteins, are shown in Figure 5A.

Tyrosine residues Tyr42 and Tyr24 are oxidized in Hb. We performed a spectrofluorimetric study of Hb tyrosine residues under ONOO^−^ impact. In metHb and Hb-DNICs, the fluorescence intensity increased with ONOO^−^ concentration growing (Figure 6A). This accounts for the formation of 3,3′-dityrosine, which has maximum fluorescence at 400 nm [43]. However, from the ratio of metHb to an oxidant of 1:10.6 (1.6 mM), the fluorescence of Hb-DNICs decreased, most likely due to 3-nitrotyrosine formation. This effect can be explained by the breakdown of most DNICs with this peroxynitrite concentration, and release of iron, which catalyzes tyrosine nitration [23].

Tryptophan is another amino acid peroxynitrite target [46]. There are six tryptophan residues in Hb, of which only Trp37β is located on the surface of the molecule. However, first of all, the Trp15β residue is oxidized, and Trp37β is oxidized only if the native protein structure is disturbed.

At the next stage, the dependence of tryptophan fluorescence at 327 nm on ONOO^−^ concentration was studied (Figure 6B). In metHb, fluorescence intensity decreased depending on the dose, indicating oxidation of tryptophan residues. In Hb-DNICs, incubated with low concentrations of the oxidizer, the fluorescence slightly increased, and decreased relative to the initial level only at the ratio of 1:17 (2.55 mM ONOO^−^) (Figure 6B, dotted line), It may be due to DNICs decay.

We cannot exclude nitrotryptophan formation in the metal-catalyzed peroxynitrite reaction with tryptophan residue. This reaction involves peroxyazotic acid or activated intermediate ONOOH*, formed from trans-peroxyazotic acid [47]. These reactions are shown on Figure 5B.

### 2.5. Oxidation of Hb Thiol Groups

In our experiments the interaction of ONOO^−^ with metHb resulted in slight oxidation of SH- groups (reactive cysteines Cys93β) (Figure 7). A ~10 % decrease in fluorescence intensity of the thiol adduct with ThioGlo1 was observed with a molar ratio of metHb:ONOO^−^ = 1:4 (0.6 mM ONOO^−^) (Figure 7, curve 1). Further increase in ONOO^−^ concentration did not lead to the increased oxidation of metHb SH-groups. 

The initial decrease of the fluorescence of thiol adduct in the sample with Hb-s (Figure 7, curve 2) indicates that the reactivity of SH-groups in relation to ThioGlo1 is reduced due to their inclusion in the complexes.

Our results are consistent with the data from literature supporting that the main target of peroxynitrite in Hb is heme, rather than Cys93β [30,48]. The rate constant for Cys93β reaction with peroxynitrite is 1.5 × 10^3^ M^−1^s^−1^ [30]. At the same time, a similar constant with the heme group of metHb ~4 × 10^4^ M^−1^s^−1^ [29,42]. It is also known that, if Hb interacts with peroxynitrite, the oxidation of tyrosine mediated by heme plays a key role in the formation of thiyl radicals of cysteine residues [42]. Therefore, phenoxyl tyrosine radicals are reduced by cysteines due to the intramolecular electron transfer. It is shown that in such processes cysteine residues that act as antiradical agents inhibit nitrotyrosine formation [49]. Along with that, Cys93β insertion into DNICs may be an additional mechanism for protecting these residues against oxidative modification [15].

### 2.6. Formation of Interprotein Cross-Links

Reduction of the oxoferryl Hb intermediate by amino-acid residues, such as tyrosine, cysteine, and tryptophan, located close to heme, leads to the formation of radicals of these amino acids. They contribute to the mostly intermolecular covalent bonds (cross-links) being formed and the protein molecules aggregated [30,50].

Inter-subunit cross-linking and protein aggregation under ONOO^−^ impact registered in SDS electrophoresis in 12% Polyacrylamide gel (PAAG) under nonreducing and reducing (with DTT) conditions. In the control variant (Hb without DNICs), a dose-dependent formation of metHb subunit dimers and high-molecular-weight aggregates took place in all the ONOO^−^ concentration range (Figure 8). With ONOO^−^ concentrations of 2.6 mM and 4.2 mM, aggregation was so intense that the protein did not even penetrate the concentrating gel. If SH groups of metHb were included into DNICs as ligands, the protein would aggregate only with ONOO^−^ concentrations of 2.6 mM (protein: ONOO^−^ molar ratio = 1:17) and 4.2 mM (1:28), and less intensively than in control (Figure 8).

The level of crosslinking of met metHb molecules did not depend on the presence of dithiotreitol, that excludes the contribution of disulfide bonds to the aggregation process. This is consistent with the data on the ONOO^−^ effect on SH groups (see Figure 7). This confirms our data on the ability of DNICs to prevent aggregation of metHb subunits under the impact of an oxidizer [15].

## 3. Discussion

In this work we studied the effect of Hb-bound DNICs on the oxidative Hb modification caused by peroxynitrite. The results obtained indicate that these complexes effectively protect Hb against oxidative degradation caused by peroxynitrite. 

The interaction of thiol and protein DNICs with peroxynitrite was previously reported [15,51]. At physiological pH, peroxynitrite oxidizes glutathione and albumin complexes by a two-electron mechanism. In the case of Hb-DNICs, this reaction looks like this: [{(CysS^−^)(R^−^)}-Fe^+^-(NO^+^)_2_]^+^ + ONOO^−^ → {(CysS^−^)(R^−^)}-Fe^+^-(ONOONO)(NO^+^) → products of DNICs degradation(7)

Reaction Equations (7), (9) and (10) describe the transformations of Hb-DNICs. Characters in curly brackets denote amino acid ligands involved in the formation of Hb-DNICs. As it was stated in the Introduction section, one of these ligands is Cys93 residue of the Hb β-subunit, and the other, denoted as R^−^, may be an anionic group of amino acid residue spatially close to Cys93. 

According to the literature data, the second cysteine of the Hb β-subunit (Cys112β) is located quite far from Cys93β [52], and can hardly be the second amino acid ligand for Hb-DNICs. Such ligands could potentially be His92β, His97β or Asp94β residues located near Cys93β [53]. Note that albumin-bound DNICs include cysteine and histidine residues, but their EPR spectra significantly differ from the spectrum of Hb-DNICs [54]. Thus, it is most likely that the ionized carboxyl group of Asp94β forms a coordination bond with the iron of Hb-DNICs and plays the role of the second amino acid ligand. However, additional studies are necessary for the final identification of the second (besides Cys93β) amino acid ligand of Hb-DNICs.

During reaction 7, unstable intermediates, nitrosyl-peroxynitrite-associated complexes, are formed [51]. The formation of complexes containing coordinated peroxynitrite is also possible [15,55]. These unstable complexes break down with the release of nitrite/nitrate, or take part in the oxidation and nitration of biomolecules, most often tyrosine residues. In Equation (7), Hb-DNICs are shown as [{(CysS^−^)(R^−^)}-Fe^+^-(NO^+^)_2_]^+^. Square brackets are used to point out the total charge of the entire complex, and to show that the total charges of both sides of this equation are equal. In the reaction Equations (9) and (10) this formula is used without square brackets.

The resistance of DNICs to peroxynitrite depends on the availability of the nitrosyl DNICs group. The glutathione DNICs complex is the most resistant to oxidation: (k = (1.8 ± 0.3) × 10^7^ M^−2^C^−1^)

The cysteine DNICs complex is oxidized more efficiently: (k = (4.0 ± 0.3) × 10^8^ M^−2^C^−1^) 
and the highest oxidation rate constant was characteristic for albumin-bound DNICs [51]:(k = (9.3 ± 0.5) × 10^9^ M^−2^C^−1^)

The cells are generally accepted to have a pool of iron, weakly bound to proteins and low-molecular-weight substances: the labile iron pool (LIP) or chelatable iron pool (CIP) [56]. LIP usually includes bivalent iron with pro-oxidant properties. However, it was shown that in macrophages, LIP/CIP are converted to DNICs during interaction with ^•^NO [57]. It is assumed that under these conditions ONOO^−^ can interact with DNICs [58]. It is possible that in RBC DNICs are formed with the participation of LIP/CIP. However, NO synthase of RBC (eNOS isoform) may be the ^•^NO source for DNICs formation [59].

It was suggested that LIP intercepts ONOO^−^ and its free radical derivatives, e.g., ^•^NO_2_ and ^•^CO_3_ [60]. The antioxidant effect of bivalent iron for peroxynitrite is explained by its ability to reduce ONOO^−^ to nitrite by two-electron mechanism, with a rate constant being two orders higher than that of ONOO^−^ reaction with CO_2_ [58,61]. LIP is believed to get converted to an oxoferryl intermediate in this reaction [58]:LIP-Fe^2+^ + ONOO^−^ → [LIP-Fe^IV^ = O]^2+^ + NO_2_^−^(8)

The oxoferryl LIP intermediate is less reactive than peroxynitrite, and it can be reduced by glutathione and ascorbate. However, oxoferryl complexes can be also reduced by ^•^NO [17,28]. 

Therefore, we can suggest another mechanism for DNICs interaction with peroxynitrite, based on the antioxidant properties of both ^•^NO and iron ions of the complex. At the first stage, a complex containing bound peroxynitrite is formed:{(CysS^−^)(R^−^)}-Fe^+^-(NO^+^)_2_ + ONOO^−^ → {(CysS^−^)(R^−^)}-Fe^2^^+^-(ONOO^−^)(^•^NO) + NO^+^(9)

This complex turned to oxoferryl intermediate, which further oxidized ^•^NO to nitrite:
{(CysS^−^)(R^−^)}-Fe^2^^+^-(ONOO^−^)(^•^NO) → {(CysS^−^)(R^−^)}-[Fe^IV^=O(^•^NO)]^2+^ + NO_2_^−^
↓
{(CysS^−^)(R^−^)} + Fe^III^ + NO_2_^−^
(10)

In turn, Fe^III^ ions contribute to homolytic cleavage of peroxynitrite and enhance the oxidation and nitration reactions due to a single-electron mechanism [41,58,62]. At the same time, under the ONOO^−^ impact, heme iron in metHb is oxidized to the oxoferryl form, porphyrin radicals are formed, and the porphyrin ring is broken. These oxidants, along with peroxynitrite and its free radical derivatives, cause oxidation of metHb amino-acid residues, including the formation of carbonyl groups.

The formation of carbonyls in proteins is enhanced in case transition metals (Fe^3+^, Cu^2+^) are present, as they could catalyze oxidation of the nearby amino-acid residues [22,63]. Taking this into account, we can assume that DNICs binding centers in a Hb molecule can also catalyze the formation of carbonyl derivatives. However, our results suggest the opposite: Hb-bound DNICs inhibited the formation of carbonyls to metHb:ONOO^−^ ratio = 1:10.6 (1.6 mM). With higher oxidant concentrations, when DNICs were decomposed with iron release, the oxidation of amino-acid residues was stimulated. The role of iron in stimulating carbonyl formation is indicated by the data of experiments with DTPA. In the presence of this chelator, the oxidative modification of metHb was almost completely suppressed in the control experiment, and in the case of Hb-DNICs inhibition, it was observed up to 1:17 mM (2.55 mM) ratio. 

Metal-containing compounds catalyze not only oxidation of the nearby amino-acid residues, but also their nitration. Transition metals catalyze nitrotyrosine formation in a bimolecular reaction with peroxynitrite [23]. Peroxynitrite-dependent nitration catalysts are Mn-superoxide dismutase (Mn-SOD), myeloperoxidase, and some hemoproteins [16,64]. Cytochrome *c* and myoglobin were shown to protect other biomolecules from irreversible modifications due to the nitration of the existing tyrosine [65]. Unlike these hemoproteins, Hb does not catalyze the nitration reaction [16]. Meanwhile, the binding centers of metals (Fe^2+^, Cu^+^, Mn^2+^) in Hb, located near tyrosine residues, provide site-specificity of their nitration [66]. Therefore, another nitrotyrosine formation pathway can be provided by DNICs [55]. Probably, DNICs are not only site-specific antioxidants, as we postulated earlier [14,15,67], but also site-specific nitrating agents. In addition, transfer of a nitrosyl fragment to protein thiols is site-specific itself [2,5,68,69,70]. The transfer of the iron-nitrosyl fragment can also be selective. 

The DNICs’ effect on proteins can be related to electron transfer between iron and ^•^NO within a [Fe^+^(NO^+^)_2_] complex, and, consequently, the appearance of nitroxyl (NO^−^) and nitrosonium (NO^+^) ions, whose reactivity is higher than that of a neutral ^•^NO molecule [1,8,41,71,72]. In particular, NO^+^ ions within DNICs react with thiol groups forming S-nitrosothiols [5]. This allows DNICs to contribute to redox signaling, which provides protection against oxidative stress. 

DNICs functions in biological systems are not limited to NO transport and deposition. They can also be selective posttranslational modifiers.

The antioxidant properties of DNICs seem paradoxical, since the complexes of redox-active metals, as noted, stimulate oxidative protein modification. However, the ability of protein-bound DNICs and low-molecular ones with thiol ligands to inhibit oxidation of biomolecules under oxidative stress was proven earlier [14,15,17,18], and one of the reasons could be the effect of the incorporation of Hb cysteine thiols into DNICs on the reactivity of these thiols [73]. The results obtained here confirm this DNICs’ ability allowing this concept to solidify further. 

It is necessary to note also that such an inducer of oxidative stress as peroxynitrite, used in our work, is known as the key tissue-damaging agent in ischemia/reperfusion [26]. Therefore, antioxidant and antiradical properties of DNICs make them promising myocardium protectors, which increase resistance to ischemia in artificial blood circulation conditions. 

Thus, Hb-bound DNICs protect Hb against peroxynitrite-caused oxidative degradation. They prevent the formation of carbonyl derivates, oxidation of tyrosine and tryptophan residues, the formation of covalent cross-links between Hb subunits and degradation of a heme group. Protective effect of DNICs can be explained by their ability to transform peroxynitrite to nonradical products and directly intercept free radicals emerged due to its homolysis. It should be noted that the complexes protect from oxidation not only the cysteine residues with which they are formed, but also the heme group of Hb. This is especially important for RBC, which contain large amounts of Hb and are constantly exposed to ROS and RNS produced in the cell and in plasma. This makes DNICs a promising substance for use as a protector of RBC during cryopreservation of blood. Another important application of these complexes may be associated with the protection of the myocardium to ischemia in cardiac arrest and reperfusion. 

## 4. Materials and Methods

### 4.1. Chemicals and Reagents

The following reagents were used in the present work: bovine methemoglobin (oxidized Hb), pyridine, sodium dithionite, glycerol, glycine, Coomassie brilliant blue R-250, 2-amino-2-(hydroxymethyl)-1,3-propanediol (Tris), 4-(2-hydroxyethyl)piperazine-1-ethanesulfonic acid (HEPES), dimethyl sulfoxide (DMSO), 2,4-dinitrophenylhydrazine (DNPH), *L*-glutathione, dithiothreitol (DTT), diethylenetriaminepentaacetic acid (DTPA), polyacrylamide (PAA), sodium dodecyl sulfate (SDS), NaNO_2_, FeSO_4_—“Sigma-Aldrich” (St. Louis, MO, USA); 10-(2,5-dihydro-2,5-dioxo-1H-pyrrol-1-yl)-9-methoxy-3-oxo-H-naphtho[2,1-b]pyran-2-carboxylic acid—methyl ester (ThioGlo1)—“Calbiochem” (Los Angeles, CA, USA); 4-hydroxy-(2,2,6,6-tetramethylpiperidin-1-yl)oxyl (4-hydroxy-TEMPO)—“Oxis” (Portland, OR, USA).

DNICs was synthesized as described in [14,15]. DNICs with phosphate ligands were obtained by passing gaseous NO in a Thunberg vessel through FeSO_4_ solution (5.5 mM) in 100 mM K,Na-phosphate buffer (pH 6.8). Hb-DNICs were obtained by adding 1 ml of 1 mM Hb solution in 0.1 M K,Na-phosphate buffer (pH 6.8) to 400 μL of 5 mM phosphate DNICs. Hb-DNICs (~1.3 mM) emerged after 5 min of incubation in the amount of ~1.8 complexes per a Hb tetramer. The DNICs concentration was determined based on the overall intensity of the electron paramagnetic resonance (EPR) signal of the complexes, with the 4-hydroxy-TEMPO spin label as an external standard. The DNICs preparations were stored at −70 °C.

Peroxynitrite was synthesized following [74], rapidly mixing 0.6 M NaNO_2_ solution and 0.6 M H_2_O_2_ solution in 0.7 M HCl. It was stabilized with 0.9 M NaOH solution. Unreacted H_2_O_2_ was removed by adding MnO_2_ powder, which was then separated by filtration. Peroxynitrite concentration was determined using a characteristic absorption band at 302 nm (ε = 1.67 mm^−1^cm^−1^).

### 4.2. Registration of EPR Spectra

EPR spectra were recorded using the E-109E spectrometer (“Varian”, Palo Alto, CA, USA) at room temperature (~25 °C). Samples (80 µL) were placed in gas-permeable Teflon capillaries PTFE 22 (“Zeus Industrial Products”, Orangeburg, SC, USA) before measurements. Conditions for recording EPR spectra: microwave power—10 mW, microwave field frequency—9.15 GHz, RF modulation amplitude—0.4 mT (for DNICs). The EPR signal of the 4-hydroxy-TEMPO spin tag was recorded under the following conditions: 10 mW microwave power, 9.14 GHz microwave field frequency, and 0.05 mT RF modulation amplitude. 

### 4.3. Determining of Protein Carbonyls

Carbonyl Hb derivatives were quantified using the method [75] with minor modifications. The method consists in inducing the formation of covalent adducts of carbonyl (aldo- and keto-) groups) with DNPH, spectrophotometrically recorded at 13 wavelengths. The amount of the formed 2,4-dinitrophenylhydrazones was calculated using the formula:S = S_1_ + S_2_, 
where
S_1_ = (E_230_ + E_254_) × 12 + (E_254_ + E_270_) × 8 + (E_270_ + E_280_) × 5 + (E_280_ + E_356_) × 38 + (E_356_ + E_363_) × 3.5 + (E_428_ + E_520_) × 46,
and

S_2_ = (E_363_ + E_370_) × 3.5 + (E_370_ + E_428_) × 29 + (E_430_ + E_434_) × 2 + (E_520_ + E_535_) × 7.5.S_1_—aldehyde-dinitrophenylhydrazones,S_2_—neutral ketone-dinitrophenylhydrazones.

The samples were prepared in the following way. Peroxynitrite was added to 0.1 mL of 0.15 mM metHb solution in 20 mM K,Na-phosphate buffer (pH 6.8) to the final concentrations of 0.38; 0.6; 1.0; 1.6; 2.55; and 4.2 mM, and incubated at room temperature for 15 min. Then 0.5 mL of DNPH solution in 2 M HCl was added to the mixture and incubated for 1 h. The protein precipitate formed after adding 0.5 mL of 20% TCA solution, and after 10 min samples were centrifugated at 3000 g for 15 min. The precipitate was thrice washed with a mixture of ethanol and ethyl acetate (1:1) (0.4 mL) to remove unbound 2,4-DNFG. The final precipitate was air-dried and dissolved in 1 mL of 8 M urea before measurement. A similar procedure was performed with the iron complex DTPA, which was added to the final 2.5 mM concentration. The samples were diluted five times prior to measurements.

Optical absorption spectra were recorded on the Cary 300 UV-VIS spectrophotometer (“VarianBio”, Palo Alto, CA, USA) at room temperature in a 1 cm optical cuvette at a scanning speed of 600 nm/min. 

### 4.4. Measurements of Heme Groups

The heme concentration in metHb solution was measured using the pyridine hemochrome method [76]; 135 µL of water and 450 µL of 30% pyridine alkaline solution were added to 15 µL of metHb sample prepared as described in Section 4.3. Before measurements, the solution was reduced with sodium dithionite. Optical absorption of the reduced heme complex with pyridine was measured at 556 and 539 nm. The heme concentration was calculated using the A_556_-A_539_, using the difference A_556_-A_539_, given (Ε_556-539_ = 4.3 mM^−1^cm^−1^).

### 4.5. Measurement of Protein SH Groups

Reduced sulfhydryl groups were quantified using the ThioGlo1 thiol-specific fluorescent label. The adding of the ThioGlo1 protein to the solution resulted in the formation of a thiol adduct with maximum emission of fluorescence emission at 500 nm at an excitation wavelength of 379 nm [77]. 

Samples were prepared as follows: 5 μL of 2.5 mM ThioGlo1 in DMSO was added to 5 μL of a reaction mixture containing 0.03 mM metHb, and incubated for 4 min. The resulting solution (10 μL) was added to a quartz spectrofluorimetric cuvette containing 490 μL of a 20 mM K-phosphate buffer (pH 6.8). The fluorescence was registered on RF-5301 PC spectrofluorimeter (“Shimadzu”, Kyoto, Japan) with high sensitivity and at a medium scanning speed (in accordance with the instrument description). The slit width was 3 nm and 5 nm for excitation and emitted light respectively. 

### 4.6. Identification of Tryptophan and Tyrosine Residues State

The state of tryptophan and tyrosine residues in the metHb molecule was studied via fluorescence spectroscopy. The tryptophan autofluorescence was selectively excited at a λ = 295 nm and recorded at 330 nm. The slit width was 5 nm for the excitation light and 10 nm for the emitted one. Dityrosine formation was detected at excitation λ = 325 nm and emission λ = 400 nm [43], the spectral range was 330–600 nm. The slit width for the excitation and emitted light was 5 nm.

The protein samples were prepared as described above in Section 4.3. Methemoglobin samples were diluted with 20 mM K-phosphate buffer (pH 6.8) 25 times for measuring tryptophan fluorescence, and 12.5 times for dityrosine one. The fluorescence was recorded in quartz cuvettes with 1 cm optical path length on RF-5301 PC spectrofluorimeter (“Shimadzu”, Kyoto, Japan) with high sensitivity and at a medium scanning speed (in accordance with the instrument description).

### 4.7. SDS-Electrophoresis in PAAG

Electrophoresis was performed in 12% PAAG blocks with a size of 15 cm × 15 cm × 1 mm according to the Laemmli method [78] with a VE-series vertical electrophoresis device (“Helicon”, Moscow, Russia). MetHb samples prepared as described above were diluted with a sample buffer with the ratio of 1:1, and heated at 95 °C for 5 min. The sample buffer contained 0.1 M Tris-HCl (pH 6.8), 4% SDS, 0.2% bromophenol blue, and 20% glycerol. 3% DTT solution was added to the buffer to ensure the reducing conditions; 10 μL of the sample was loaded to the gel. The electrode buffer contained 0.2 M Tris-glycine (pH 8.3), and 0.1% SDS. 

Electrophoresis was carried out at 4 °C, I = 50 mA, and U = 150 V. Elf-4 (“DNA-Technology”, Moscow, Russia) was used as a power source for electrophoresis. After completing protein separation, the gel was fixed and stained with Coomassie brilliant blue R-250 solution.

### 4.8. Statistical Analysis

The measurements were performed in at least three replicates for each sample. The statistical data were processed based on 3–4 analytical repetitions. The data are presented as a mean ± standard deviation. Statistical differences were determined by means of a one-way ANOVA analysis followed by post hoc Tukey’s multiple comparison test. The differences were considered to be statistically significant when the *p* value was less than 0.05.

## Figures and Tables

**Figure 1 ijms-22-13649-f001:**
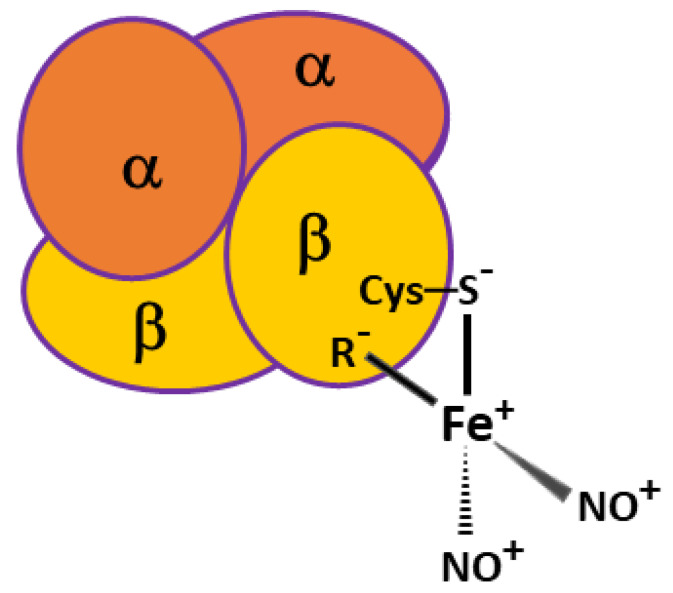
Schematic image of dinitrosyl iron complex (DNIC) bound with tetrameric erythrocytic hemoglobin (Hb-DNIC).

**Figure 2 ijms-22-13649-f002:**
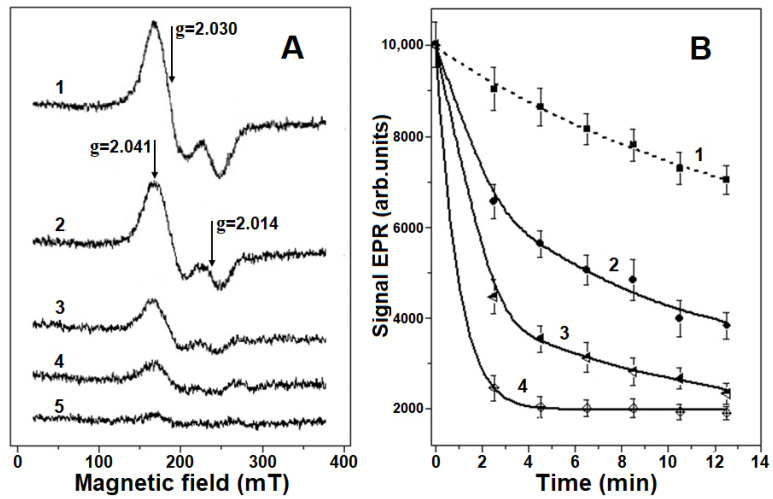
Destruction of metHb-bound dinitrosyl iron complexes (Hb-DNICs) with peroxynitrite (ONOO^−^). The reaction mixture contained 0.2 M Na, K-phosphate buffer (pH 7.2) and 0.4 mM Hb-DNICs (Hb concentration). (**A**)—typical electron paramagnetic resonance (EPR) spectra recorded 4.5 min after adding various peroxynitrite concentrations to the reaction mixture: 1—Hb-DNICs without peroxynitrite, 2—Hb-DNICs + 1.25 mM (1:3.2), 3—Hb-DNICs + 2.5 mM ONOO^−^ (1:6.25), 4—Hb-DNICs + 4.8 mM ONOO^−^ (1:12), 5—Hb-DNICs + 5.8 mM ONOO^−^ (1:14.5), the molar ratio of Hb or Hb-DNICs to ONOO^−^ is shown in parenthesis. (**B**)—kinetics of Hb-DNICs degradation with the action of various ONOO^−^ concentrations (results of five experiments).

**Figure 3 ijms-22-13649-f003:**
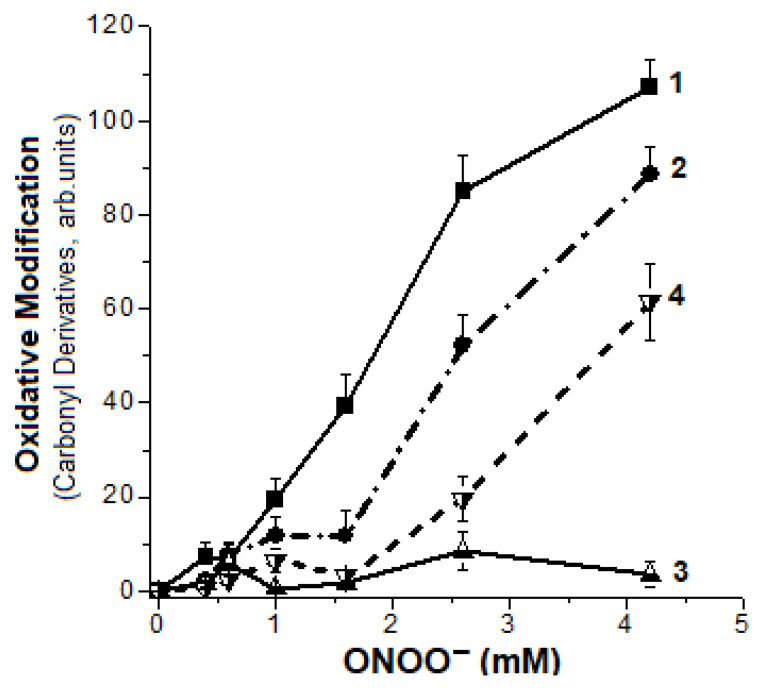
Formation of metHb carbonyl derivatives under the impact of various concentrations of peroxynitrite. Reaction mixture contained—0.2 M Na,K-phosphate buffer (pH 6.8) and 0.15 mM Hb-DNICs (Hb concentration). Curves 1 and 3metHb, curves 2 and 4—Hb-DNICs, curves 3 and 4—with DTPA. The modification was carried out at the following concentrations of ONOO^–^ in the reaction medium: 0 mM, 0.38 mM (1:2.5), 0.6 mM (1:4), 1 mM (1:6.5), 1.6 mM (1:10.6), 2.55 mM (1:17), 4.2 mM (1:28). The molar ratio of metHb or Hb-DNICs to ONOO^-^ is shown in parentheses.

**Figure 4 ijms-22-13649-f004:**
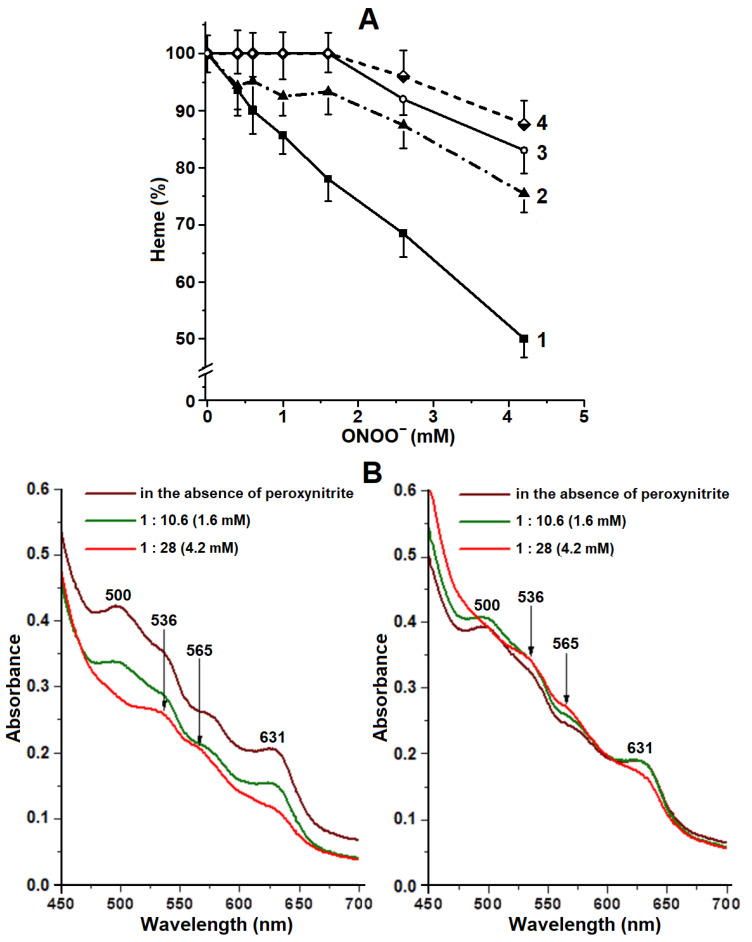
(**A**)—destruction of the heme group of methemoglobin under the impact of peroxynitrite. Curves 1 and 3—metHb, curves 2 and 4—Hb-DNICs, curves 3 and 4—with diethylenetriaminepentaacetic acid (DTPA). The composition of the reaction medium, including peroxynitrite concentration, is the same as in the legend to Figure 3. (**B**)—Changes in the state of the heme group in Hb-DNICs during the oxidation by peroxynitrite. Left—metHb, right—Hb-DNICs.

**Figure 5 ijms-22-13649-f005:**
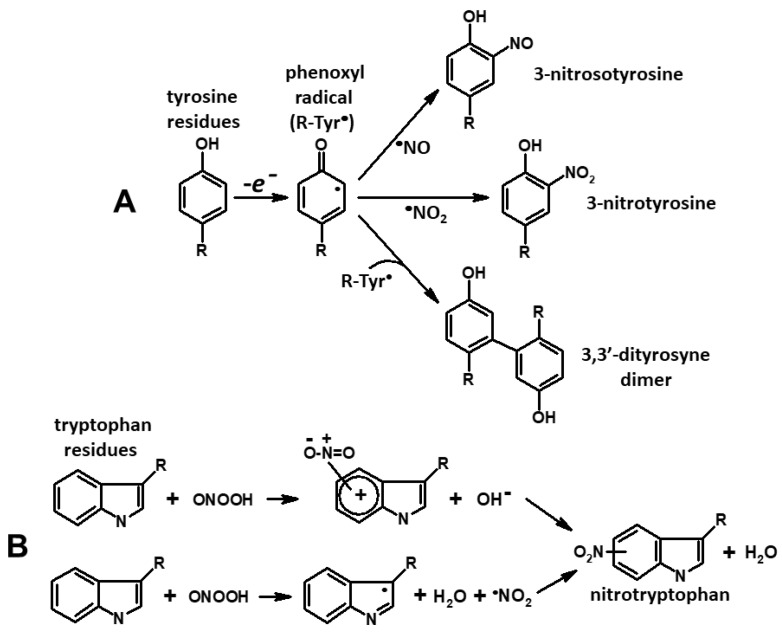
Pathways of nitration of tyrosine (**A**) and tryptophan residues (**B**). Scheme **A** also shows the formation of 3,3′-dityrosine dimer and 3-nitrosotyrosine.

**Figure 6 ijms-22-13649-f006:**
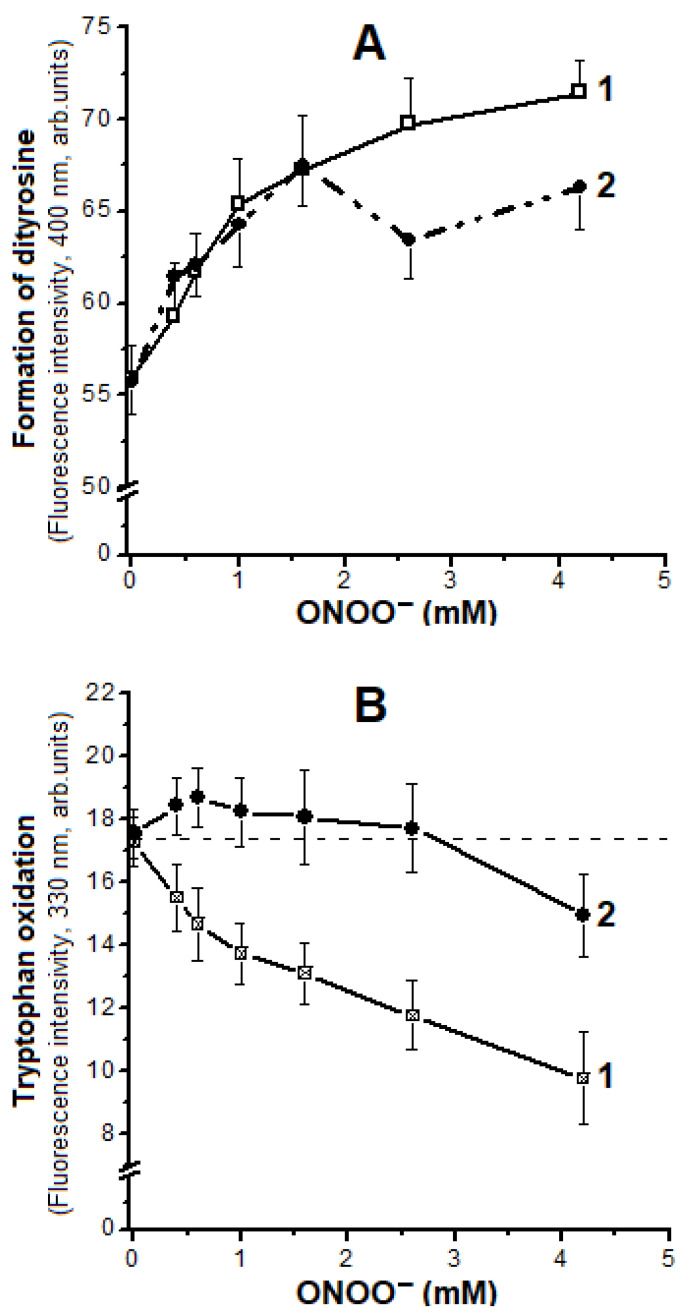
Changes in fluorescence during the formation of dityrosines (**A**), and during the oxidation of tryptophan residues (**B**) under peroxynitrite action. Curve 1—metHb, 2—Hb-DNICs. The composition of the reaction medium including peroxynitrite concentration is the same as in the legend to Figure 3.

**Figure 7 ijms-22-13649-f007:**
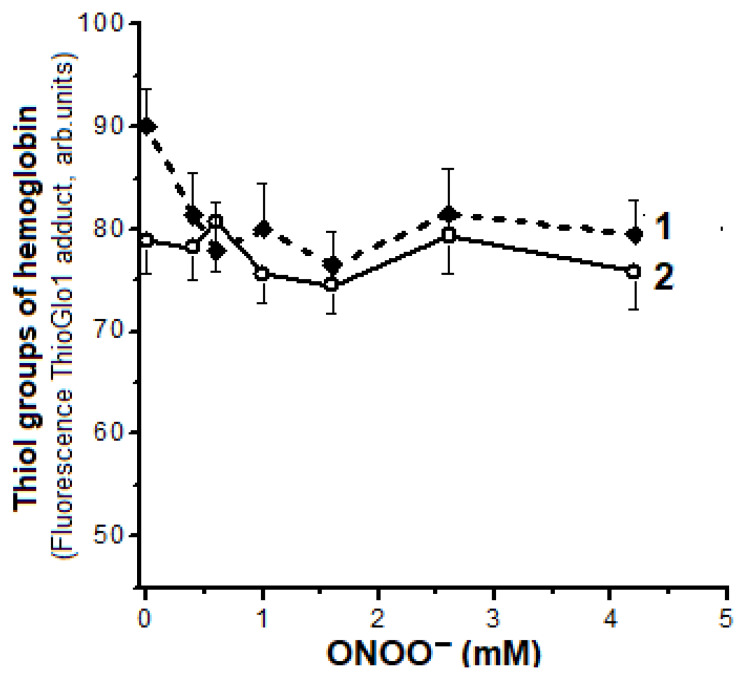
Influence of ONOO^−^ on the amount of SH- groups of metHb detected by 10-(2,5-dihydro-2,5-dioxo-1H-pyrrol-1-yl)-9-methoxy-3-oxo-H-naphtho[2,1-b]pyran-2-carboxylic acid–methyl ester (ThioGlo1), which forms an adduct with maximum fluorescence at 500 nm at an excitation wavelength of 379 nm. The ratio of metHb to ThioGlo1 in the reaction medium is 1:8 mM. Curve 1—metHb, 2—Hb-DNICs. The composition of the reaction medium, including ONOO^−^ concentration, is the same as in the legend to Figure 3.

**Figure 8 ijms-22-13649-f008:**
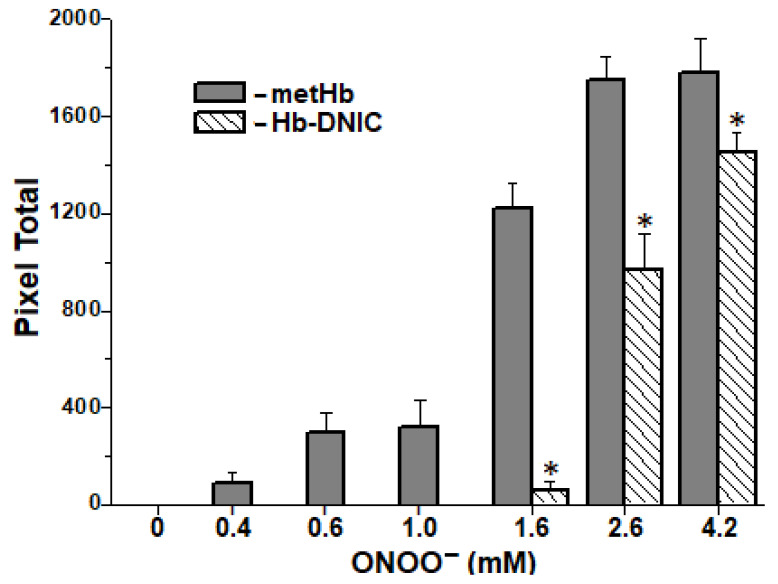
Total amount of covalent aggregates of metHb subunits. According to the data of SDS electrophoresis in 12 % PAAG with dithiothreitol (DTT). * Differences were considered significant at *p* < 0.05.

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
