# Peer review of "Protective Effect of Dinitrosyl Iron Complexes Bound with Hemoglobin on Oxidative Modification by Peroxynitrite"

_ijms, 2021, doi:10.3390/ijms222413649_

Round 1

Reviewer 1 Report

I recommend the publication of the manuscript in its current form.

"Though crucial to understanding degradation, the protection of Hb bestowed by Hb bound DNICs has not been thoroughly studied in the literature. Authors in this manuscript have demonstrated that Hb-DNICs are destroyed under the peroxynitrite action as revealed by EPR spectroscopy. It was also found that they prevent the formation of crosslinks between different Hb subunits. Overall, the study sheds new light on the protection ability of DNICs and their ability to transform peroxynitrite to non-radical products. The data in the manuscript support the claims. I recommend the publication of this manuscript in the present form"

Author Response

We want to thank eminent reviewer for hard job of reviewing our manuscript and for warm words about it. It will help us to continue our research in this field.

Reviewer 2 Report

In this work, authors have seen the protective Effect of dinitrosyl iron complexes bound with hemoglobin on oxidative modification by peroxynitrite. The work is appropriate for publication in IJMS, however, I would suggest the authors to upgrade the introduction section in which the importance of the work should be elaborated. 

Author Response

We want to thank eminent reviewer for hard job of reviewing our manuscript, for warm words about it, and for useful comments.

According to reviewers advice, the introduction section of the manuscript was edited to make aims and goals of the work clearer, and better relate them to the results obtained. 

Author Response

We want to thank the eminent reviewer for hard job of reviewing our manuscript and for important and useful comments made. We took into account almost all comments and made appropriate changes to the manuscript.

Please see the attachment where we have written our answers to the reviewer's comments.

Round 2

Reviewer 3 Report

The current version was almost considered by my reviewer comments.

However, I found more questionable points, therefore, you have to take care of them before the next step.

Introduction

You said as “a common formula [(RS)2–Fe+–(NO+)2]”. However, the charge of this formula is “+”. Even if you can find the literature formula, it is wrong. It should be “[(RS)2–Fe+–(NO+)2]+”. Or (I do not like this formula) you did not note any charges, so you should delete “[]” as (RS)2–Fe+–(NO+)2.

Discussion

You noted as “The corresponding changes are introduced into the formulas in the proposed reaction equations, for example:

{(CysS)(R)}-Fe+-(NO+)2 + ONOO→ {(CysS)(R)}-Fe+-(ONOONO)(NO+) (7).”

I commented before, what is “R”?

And the left formula is “+”, so you should note as “[{(CysS)(R)}-Fe+-(NO+)2]+”.

In this case, the total charges of both side of this equation is different, –1 in left side and neutral in right side. It is strange.

Author Response

We thank the eminent reviewer for the benevolent attitude to our answer for his previous comments and for the continuation of the hard job on reviewing our manuscript. All these comments really help us to make our manuscript better.

We hope that new changes made to our manuscript will meet the requirements of the eminent reviewer.
